DATA RELEASE

# Efficient and stable metabarcoding sequencing data using a DNBSEQ-G400 sequencer validated by comprehensive community analyses

Xiaohuan Sun[1], Yue-Hua Hu[2,*], Jingjing Wang[3], Chao Fang[1,4], Jiguang Li[3], Mo Han[1], Xiaofang Wei[3], Haotian Zheng[1,5], Xiaoqing Luo[1,6], Yangyang Jia[1], Meihua Gong[3], Liang Xiao[1] and Zewei Song[1,*]

1  BGI-Shenzhen, Shenzhen 518083, China
2  CAS Key Laboratory of Tropical Forest Ecology, Xishuangbanna Tropical Botanical Garden, Chinese Academy of Sciences, Mengla 666303, China
3  MGI, BGI-Shenzhen, Shenzhen 518083, China
4  Shenzhen Key Laboratory of Human Commensal Microorganisms and Health Research, BGI-Shenzhen, Shenzhen 518083, China
5  BGI Education Center, University of Chinese Academy of Sciences, Shenzhen 518083, China
6  State Key Laboratory of Biocontrol and Guangdong Provincial Key Laboratory of Plant Resources, School of Life Sciences, Sun Yat-Sen University, Guangzhou 510275, China

**Submitted:** 22 December 2020

\* Corresponding authors. E-mail: huyuehua@xtbg.org.cn; songzewei@cngb.org

Preprint submitted at https://doi.org/10.1101/2020.07.02.185710

## ABSTRACT

Metabarcoding is a widely used method for fast characterization of microbial communities in complex environmental samples. However, the selction of sequencing platform can have a noticeable effect on the estimated community composition. Here, we evaluated the metabarcoding performance of a DNBSEQ-G400 sequencer developed by MGI Tech using 16S and internal transcribed spacer (ITS) markers to investigate bacterial and fungal mock communities, as well as the ITS2 marker to investigate the fungal community of 1144 soil samples, with additional technical replicates. We show that highly accurate sequencing of bacterial and fungal communities is achievable using DNBSEQ-G400. Measures of diversity and correlation from soil metabarcoding showed that the results correlated highly with those of different machines of the same model, as well as between different sequencing modes (single-end 400 bp and paired-end 200 bp). Moderate, but significant differences were observed between results produced with different sequencing platforms (DNBSEQ-G400 and MiSeq); however, the highest differences can be caused by selecting different primer pairs for PCR amplification of taxonomic markers. These differences suggested that care is needed while jointly analyzing metabarcoding data from differenet experiments. This study demonstrated the high performance and accuracy of DNBSEQ-G400 for short-read metabarcoding of microbial communities. Our study also produced datasets to allow further investigation of microbial diversity.

**Subjects**  Genetics and Genomics, Metagenomics, Microbial Ecology

## DATA DESCRIPTION

We examined the accuracy of DNBSEQ-G400 sequencing on bacterial and fungal mock samples, and on a large set of soil fungal communities, and produced a dataset suitable for

benchmarking and comparison studies. Independently sequenced PCR and sequencing run technical replicates were used to examine the performance of DNBSEQ-G400's single-end 400 bp (SE400) and paired-end 200 bp (PE200) sequencing modes. Moreover, to assess the performance consistency by a single sequencing mode within one platform, all soil samples were sequenced in SE400 mode three times. Finally, to validate these data, we compared the performance in measuring the alpha and beta diversity of the soil fungal communities between two sequencing platforms, using DNBSEQ-G400 and Illumina MiSeq.

## CONTEXT

Culture independent studies have greatly expanded our knowledge of microbial diversity in recent decades [1–3]. Studies focused on characterizing microbial community composition often amplify the 16S or internal transcribed spacer (ITS) regions of rRNA [4, 5], which are the most commonly used barcoding regions for bacteria and fungi, respectively. High-throughput sequencing (HTS)-based metabarcoding allows the parallel analysis of large numbers of samples from different natural environments, especially those with complex taxonomic composition or low microbial biomass, such as soil [6], air [7], glaciers [8], and the deep sea [9].

Factors contributing to ecological biases in HTS-based metabarcoding have been reported in many studies, based on factors occurring both before (owing to marker gene length, AT/GC bias, the choice of PCR polymerase, primers and number of cycles [10–12]), and after sequencing (owing to choice of data-analyzing methods [13–15]). In addition, using mock communities, run-to-run variation in taxon presence and abundance was also observed among 16S HTS using Illumina HiSeq [16]. Similarly, bias among sequencing runs in detecting fungal taxa in soil samples was also reported [17], indicating the necessity of using positive controls in each sequencing study. Compared to the intensive focus on biases in pre- and post-HTS processes, relatively few studies have focused on the potential bias introduced by sequencers and sequencing modes.

HTS datasets of soil microbial communities can be produced by various sequencing platforms, including 454 (a platform no longer in use), IonTorrent, Pacific Biosciences (PacBio), Oxford Nanopore Technologies (ONT), and – the most frequently used – Illumina [6, 18]. In 2015, the DNBSEQ™ platform was released by MGI Tech., using combinatorial Probe-Anchor Synthesis (cPAS) and DNA Nanoball (DNB) techniques [19]. Since then, the DNBSEQ™ platform has been applied in various areas of genomics [19–21] and metagenomics [22, 23], producing results of highly comparable quality to other platforms [24, 25]. In 2017, an ultra-HTS sequencer, DNBSEQ-G400 (formerly known as MGISEQ-2000), which supported 2 × 200 paired-end and 400 single-end data, with a maximum throughput of 720 Gbp, was launched by MGI Tech. In comparison, Illumina's MiSeq and NovaSeq are capable of generating 2 × 300 and 2 × 250 paired-end data, have a 15 Gbp and 400 Gbp maximum throughput. With the comparable (and growing) read length and larger throughput, DNBSEQ-G400 can be seen as an attractive and highly capable alternative for microbial HTS analyses. Studies comparing both types of platforms are thus needed to reveal possible important differences between them and to support informed decision-making when selecting the sequencing strategy.

## METHODS

A protocol collection gathering together methods for amplification, library preparation and sequencing at DNBSEQ-G400 and Illumina MiSeq is available at protocols.io (Figure 1 [26]).

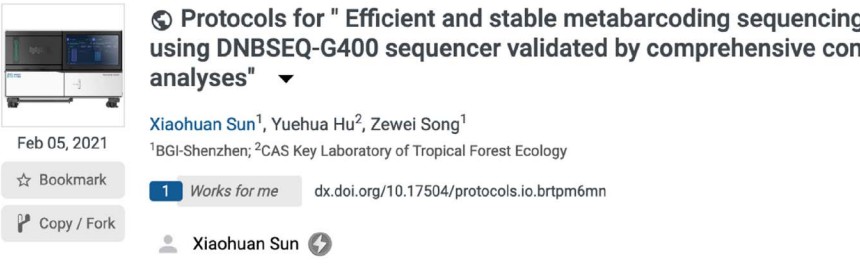

Feb 05, 2021

**Protocols for " Efficient and stable metabarcoding sequencing data using DNBSEQ-G400 sequencer validated by comprehensive community analyses"** ▾

Xiaohuan Sun[1], Yuehua Hu[2], Zewei Song[1]
[1]BGI-Shenzhen; [2]CAS Key Laboratory of Tropical Forest Ecology

☆ Bookmark

⌥ Copy / Fork

1 Works for me    dx.doi.org/10.17504/protocols.io.brtpm6mn

👤 Xiaohuan Sun ⚡

**Figure 1.** Protocol collection for efficient and stable metabarcoding using the DNBSEQ-G400 sequencer and validated by comprehensive community analyses. https://www.protocols.io/widgets/doi?uri=dx.doi.org/10.17504/protocols.io.brtpm6mn

## Sampling and sequencing experimental design

Two mock communities containing bacterial and fungal strains, respectively, were used in our study. ZymoBIOMICS Microbial Community DNA Standard D6305 was purchased from ZymoBIOMICS™; the 16S V4 region was examined for Zymo mock. Strains for fungal mock community standard (equally mixed genomic DNA of *Aspergillus ustus*, *Trichoderma koningii*, *Penicillium expansum*, *Aspergillus nidulans*, *Penicillium chrysogenum*, *Trichoderma reesei* and *Trichoderma longibrachiatum*) were obtained from China General Microbiological Culture Collection Center; the ITS2 region was examined for this mock. Taxonomic identification of the strains was confirmed by Sanger sequencing of full-length rRNA [27]. The genomic DNA of *Lentinula edodes* was mixed with gDNA of the fungi *Flammulina velutipes*, *Pleurotus eryngii* and *Saccharomyces cerevisiae* (obtained as commercial products intended for human consumption), in ratios of 1:1, 1:10 and 1:100, respectively. Meanwhile, the ITS2 PCR product of *L. edodes* was mixed with the ITS2 PCR products of *F. velutipes*, *P. eryngii* and *S. cerevisiae*, in ratios of 1:1, 1:10 and 1:100, respectively, to determine sequencing sensitivity.

A total of 1276 topsoil samples (described in GigaDB [27]) were collected from Nabanhe (NBH) 20 ha (400 × 500 m) tropical rainforest plot, Ailaoshan (ALS) 20 ha (500 × 400 m) subtropical evergreen forest plot, and Lijiang (LJ) 25 ha (500 × 500 m) subalpine forest plot, Yunnan, southwest China. We collected topsoil cores at each site (0–10 cm in depth) by hammering a ring knife (10 cm in diameter) into the soil at a regular grid of points every 50 m. Each base sample was paired with three additional samples at three random selected distances out of 0.2 m, 0.5 m, 2.5 m, 5 m, 10 m, 15 m or 24.9 m, in random compass directions, to capture variation in soil fungal composition at finer scales. Before sampling, all litter and loose debris above the sample points was removed from the forest floor. As a result, 396, 396 and 484 soil cores were collected from NBH, ALS and LJ, respectively. Soil samples were immediately stored on ice during collection, then 0.5 g soil was weighed for DNA extraction. Soil samples were stored at −80 °C after collection, and DNA from these samples was extracted within 2 months.

## DNA extraction

DNA extractions of soil samples and all fungal strains in this study were performed using the PowerSoil®DNA Isolation Kit (Mobio) according to the manufacturer's instructions. All soil samples were extracted once. DNA concentration was measured using Qubit 3.0 (Invitrogen).

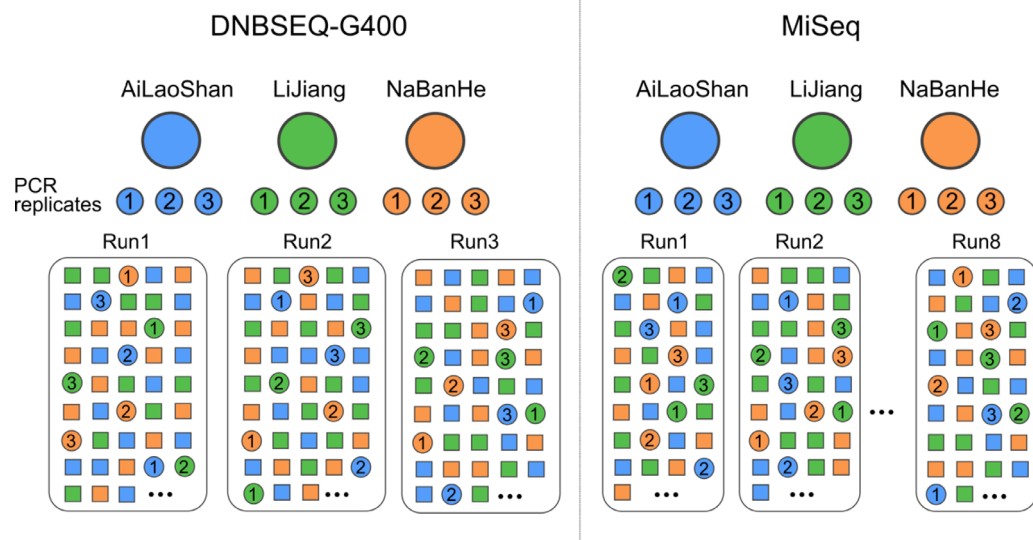

**Figure 2.** Schematic representation of sequencing strategy at DNBSEQ-G400 platform and Illumina MiSeq platform. Filled squares indicate 1276 topsoil samples for three forest plots, and filled circles indicate the technical replicates in each sequencing run. PCR replicates were labeled by the number in each filled circle.

## Amplification, library preparation and sequencing with DNBSEQ-G400 and Illumina MiSeq

Protocols are available at protocols.io [26] and outline the following. We amplified the fungal ITS1 region with primer pairs ITS1Fngs (GGTCATTTAGAGGAAGTAA) and ITS2ngs (TTYRCKRCGTTCTTCATCG); the ITS2 region with with forward primer gITS7ngs (GTGARTCATCRARTYTTTG) and reverse primer ITS4ngsUni (CCTSCSCTTANTDATATGC). All amplifications were performed with Kapa Hifi DNA polymerase (Roche).

For soil samples, ITS2 amplicons were randomly and equally distributed into three DNBSEQ runs for both 2 × 200 bp paired-end (PE200) and 400 bp single-end sequencing (SE400) (Figure 2). The left libraries of 1276 samples were repetitively sequenced two more times by SE400 using different DNBSEQ-G400 machines. Three randomly chosen soil samples from each forest plot (ALS268, LJ105 and NBH217) were all separately amplified three times as PCR replicates. The corresponding nine sequencing libraries were sequenced repetitively by three DNBSEQ runs. Sequencing was carried out using the entire lane of the Illumina MiSeq platform at the University of Minnesota Genomic Center (Figure 2). Amplicons of the 1276 forest soil samples were normalized and pooled randomly in equal molar ratios into eight parallel sequencing runs on the MiSeq platform. Again, soil samples (ALS268, LJ105 and NBH217) were all separately amplified three times as PCR replicates. The corresponding nine sequencing libraries were sequenced repetitively by eight MiSeq runs.

## Metabarcoding data processing

On average, 688.9 million high-quality reads were produced per lane on the DNBSEQ platform using 2 × 200 paired-end mode; 88.2% of bases had a Phred score of at least Q30 for read1, and 86.6% bases had a Phred score of at least Q30 for read2. After barcode demultiplexing, an average of 558,000 reads per sample were obtained. For 400 bp

single-end mode in DNBSEQ, an average of 452.5 million high-quality reads per lane were produced, with 79.4% of bases scoring Q30. An average 736,000 reads per sample were obtained after barcode splitting.

All sequencing data were processed using a Python-based Snakemake pipeline [28]. For paired-end sequencing data (PE200) from DNBSEQ-G400, the sequencing adapter and the primer area were removed by cutadapt (cutadapt, RRID:SCR_011841) [29]. A 42-bp tail representing the large subunit (LSU) was removed from the end of the reverse primer, so the entire amplicon could be aligned globally to the reference. For the DNBSEQ library, about half of the amplicons were sequenced from the 5′ end of the forward primer, and the rest from the 5′ end of the reverse primer (similar to Bahram *et al.* [6]). These two types of amplicons were separated by recognizing the leading primers using cutadapt, then corrected to the same orientation using seqtk (Seqtk, RRID:SCR_018927) [30]. Low quality reads with a maximum expected error rate >1 and minimum lengths shorter than 100 bp were discarded using Vsearch –fastq_filter function [31]. R1 and R2 reads passing these quality filtering steps were aligned to SILVA (v132) [32] or UNITE (v7.2) databases [33] using BURST v0.99.8 [34], with a 97% sequence similarity threshold in BEST mode (only the first highest hit was reported for each query). Alignment results from R1 and R2 reads were compared to ensure this target continued to be met. Operational taxonomic unit (OTU) counts were calculated based on alignment results for the community profile.

Bioinformatic analysis was slightly modified for single-end sequencing data (SE400) from DNBSEQ-G400. Like PE200 data, reads were corrected to the same orientation by recognizing the leading primers using cutadapt. Reads were trimmed until they reached the maximum expected rate ≤1 (–truncee 1). Reads shorter than 100 bp were then discarded. Community profiles were calculated using the same method as the PE200 data by aligning to the UNITE database.

Reads from MiSeq in PE300 mode were processed with a slightly modified method [28]. Briefly, 41 bp was cut from the 5′ end of the reverse primer to remove the LSU region, owing to the 1-bp primer shift. The same quality control and alignment rules were conducted for MiSeq, similar to that described above. Community profiles were calculated using identical parameters as DNBSEQ-G400 PE200 data.

Sequencing data from both MiSeq and DNBSEQ-G400 was normalized to 5000 sequences per sample by picking the sampling result with median richness from 1000 independent rarefactions. Finally, samples with less than 5000 reads were filtered out. Ultimately, 1220 soil samples remained for MiSeq, 1182 for DNBSEQ-G400 PE200, and 1202, 1195, 1183 samples remained for the three sequencing replicates by SE400 modes. The remaining 1144 soil samples across all sequencing platforms were used in all further analyses.

## Evaluation of the quality of DNBSEQ-G400 and MiSeq sequencing data

A heterogeneity G-test of goodness-of-fit and a pooled G-test of goodness-of-fit were performed using the RVAideMemoire package in R (RVAideMemoire, RRID:SCR_015657) [35]. Procrustes analysis was performed using the Procrustes and protest functions from the vegan package in R (vegan, RRID:SCR_011950) [36]. Correlation coefficients and statistics were calculated using the Procrustes permutation test in R [37]. Nonmetric multidimensional scaling (NMDS) analysis was performed using the metaMDS function from the vegan package in R [38]. Permutational multivariate analysis of variance

(PERMANOVA) was performed on the abundance of OTUs using Bray–Curtis distance with the adonis function from the vegan package in R [39]. Significant differences in abundance among major OTUs were examined using the Kruskal–Wallis test in R [37]. All data were represented by ggplot2 in R (ggplot2, RRID:SCR_014601) [40]. All R scripts can be found in GitHub [28].

## DATA VALIDATION AND QUALITY CONTROL

### The accuracy of amplicon sequencing on DNBSEQ-G400

The performance of the DNBSEQ-G400 for HTS was first examined by 16S V4 amplicons of the broadly used ZymoBIOMICS Microbial Community Standard. The proportions of each taxon in the Zymo mock community were well reproduced (Figure 3a), with no significant difference in the theoretical mock community relative abundance detected by heterogeneity G-test of goodness-of-fit ($P$ = 0.98) and pooled G-test of goodness-of-fit ($P$ = 1).

An even mock community with seven common fungal species belonging to the same genus was used to assess the classification resolution at species level by DNBSEQ-G400 (Figure 3b). The ITS2 region of the seven species was identified. Some species were assigned to the correct genus but others, for example *A. nidulans*, was identified as *A. quadrilineatus*, and *P. expansum* was identified as *P. marinum*. This was a consequence of identical ITS2 sequences between these species in the UNITE database. Thus, the resolution of the ITS2 marker allowed taxonomic assignments at genus level to be as accurate as possible (Figure 3b). The abundances of five fungi were slightly underestimated, and the overestimation of *T. longibrachiatum* was about 2-fold (Figure 3b).

Three paired combinations in geometric series (1:1, 1:10, and 1:100) of fungal taxa from ITS1 amplicons were also used to evaluate the performance of DNBSEQ-G400 (Figure 3c). Genomic DNA usually contains an undefined number of rRNA copies, so to quantify the accuracy between the amount of input and the results obtained by sequencing, we compared mixtures with the same geometric series, using both raw genomic DNA and PCR products. Generally, PCR product mixtures reproduced the original proportions better ($P$ = 0.18 by heterogeneity G-test of goodness-of-fit, and $P$ = 0.15 by pooled G-test of goodness-of-fit) than raw genomic DNA ($P$ = 0.00 by heterogeneity G-test of goodness-of-fit) (Figure 3c). This was logical since preferential PCR amplification of certain templates skewed the representation of taxa with amplification with prior-mixed DNA. Two pairs of combinations, *L. edodes* with *F. velutipes* and *L. edodes* with *P. eryngii*, were generally restored better than with the *L. edodes* and *S. cerevisiae* combination (Figure 3c). More specifically, *L. edodes* was overestimated (Figure 3c).

### Reproducibility and precision of two DNBSEQ-G400 sequencing modes

ITS2 amplicons from forest soil samples were examined using PE200 and SE400 sequencing modes of DNBSEQ-G400 (Figure 4). Thirty-three classes were identified by both modes from the ALS forest plot. Forty-two and forty-three fungal classes were detected by PE200 and SE400, respectively, from the LJ forest plot, of which 40 were shared by both modes. Finally, from the NBH plot, 42 overlapped classes were found by both modes, and 43 and 45 fungal genera were identified by PE200 and SE400, respectively. Heatmap analysis showed the frequency of occurrence of classes remained similar across the sequencing platforms,



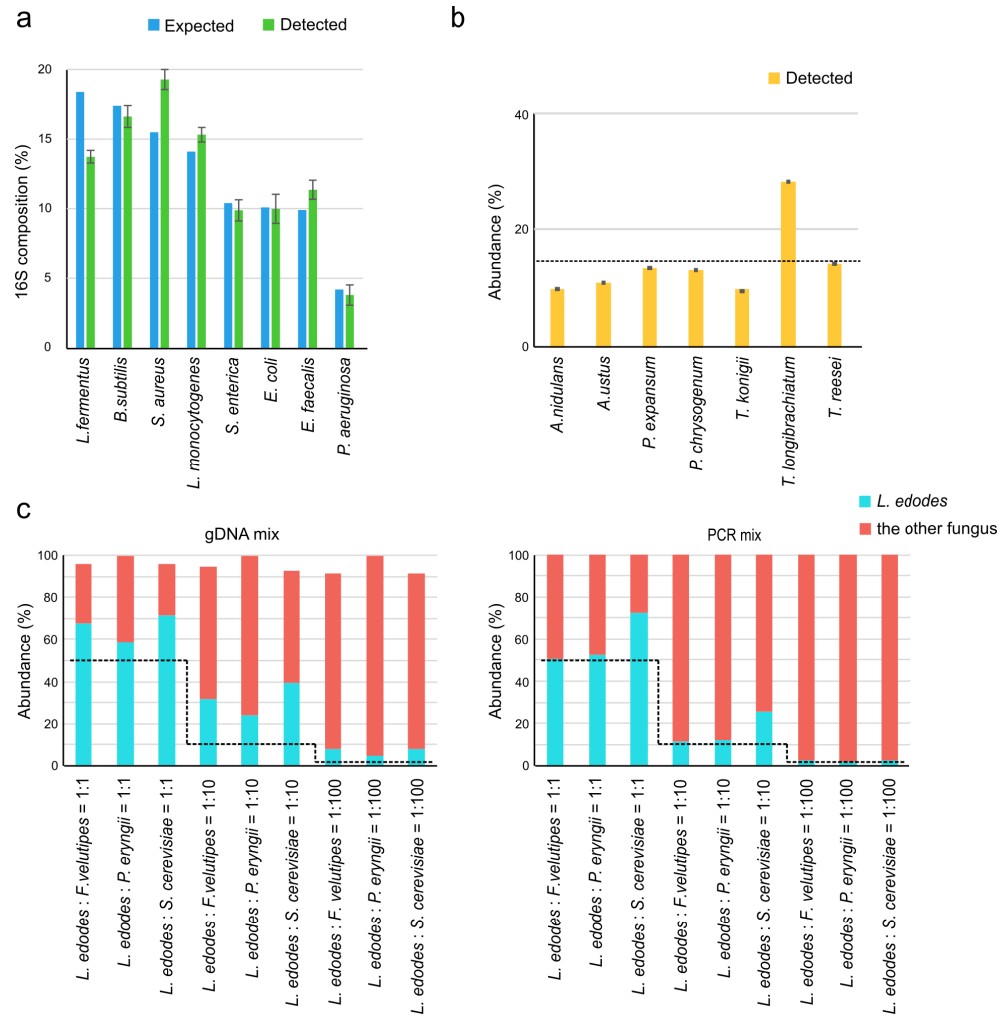

**Figure 3.** Histograms showed compositions of mock communities detected by DNBSEQ-G400 PE200 mode, (a) ZymoBIOMICS Microbial Community Standards, (b) equal genomic DNA mixtures of seven fungi mock, (c) Fungus mixed with differential ratios. The expected ratios were shown with dotted lines for b and c. Error bars indicate the standard deviation.

especially for the top 20 most abundant classes (Figure 4a). SE400 has higher sensitivity since it recovered a few more fungal classes (GS18, Kickxellomycetes, Paraglomeromycetes and Pucciniomycetes); however, PE200 performed better on the ALS and LJ samples, with more recovered classes of Zoopagomycetes and Archaeorhizomycetes. More variable classes, such as the Kickxellomycetes, Entorrhizomycetes, and Entomophthoromycetes, were found in low abundance (Figure 4a).

We next compared the two sequencing modes at the genus level; most detected genera overlapped within PE200 and SE400 (Figure 4b). Most of the fungal genera that differed between the two modes were the low abundance ones (genera abundance <0.1%), with a few exceptions. *Archaeorhizomyces* was the only genus highly detected by PE200 and barely detected by SE400. A few more genera were more abundantly detected by SE400 than

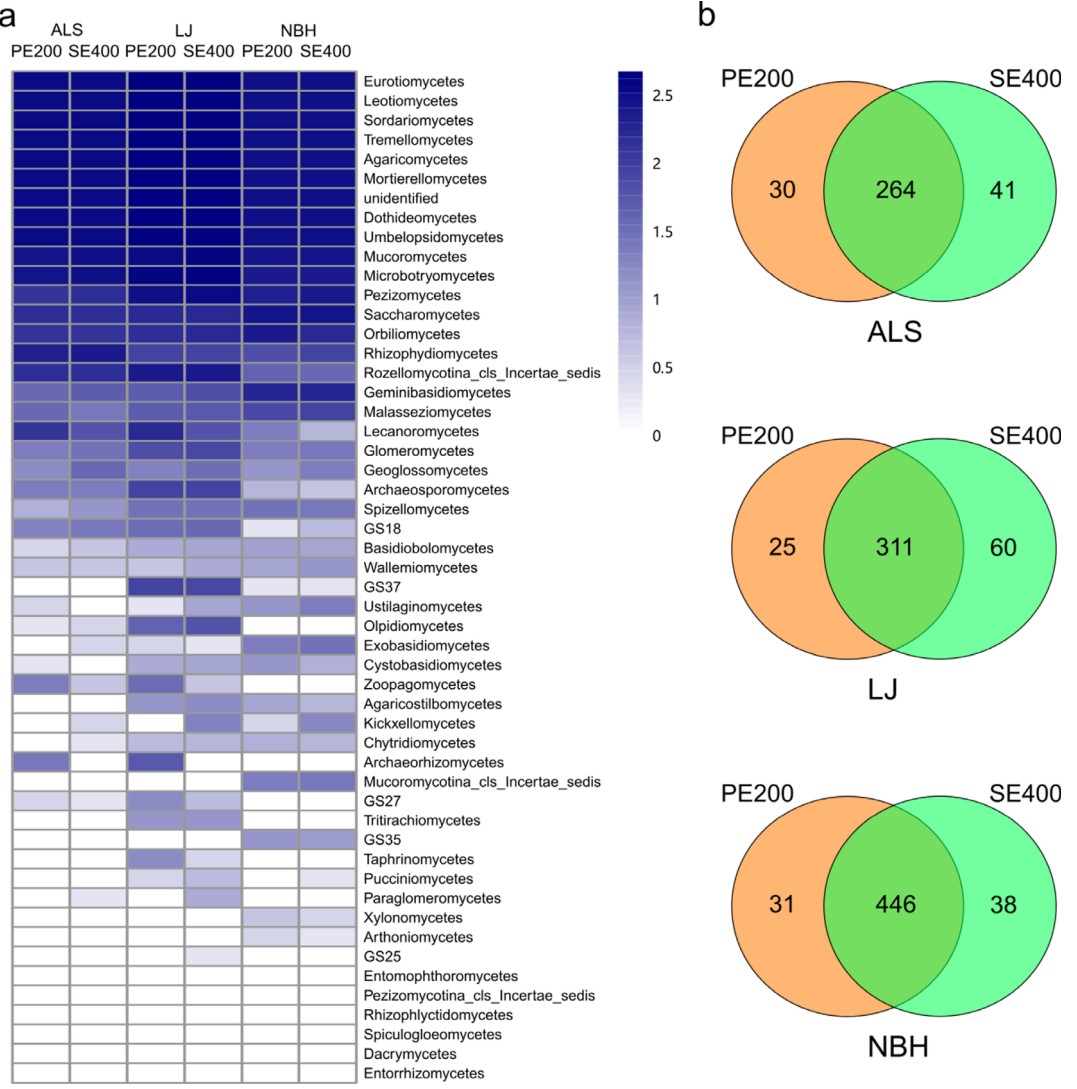

**Figure 4.** (a) All classes from the three forest plots detected by DNBSEQ-G400 are represented as log $_{10}$ (frequency) by heatmap. (b) Venn diagrams show the overlap of identified fungal genera by DNBSEQ-G400 PE200 and SE400 modes. ALS = AiLaoShan, LJ = LiJiang, NBH = NaBanHe.

PE200, including *Alpinaria, Purpureocillium, Cladorrhinum, Didymella, Talaromyces, Phoma, Helicodendron, Volucrispora, Stagonosporopsis.*

Sample-to-sample comparisons allowed us to further quantify the proportion difference of detected OTUs by the two sequencing modes on a global scale. We estimated the similarity of soil taxa beta-diversity between PE200 and SE400 using Procrustes analysis [41]. The two sequencing modes resulted in a high correlation coefficient among microbial taxa using the Procrustes permutation test (ALS = 0.948, LJ = 0.911, NBH = 0.867). These high similarities support the view that our findings are real and independent of the possible methodological biases introduced by the different sequencing modes.

To further examine the sequencing reproducibility of DNBSEQ-G400, the same library of all soil samples was sequenced in triplicate in SE400 mode (replicates SE400_1, SE400_2, SE400_3), with the second and third replicate sequenced on a different DNBSEQ-G400

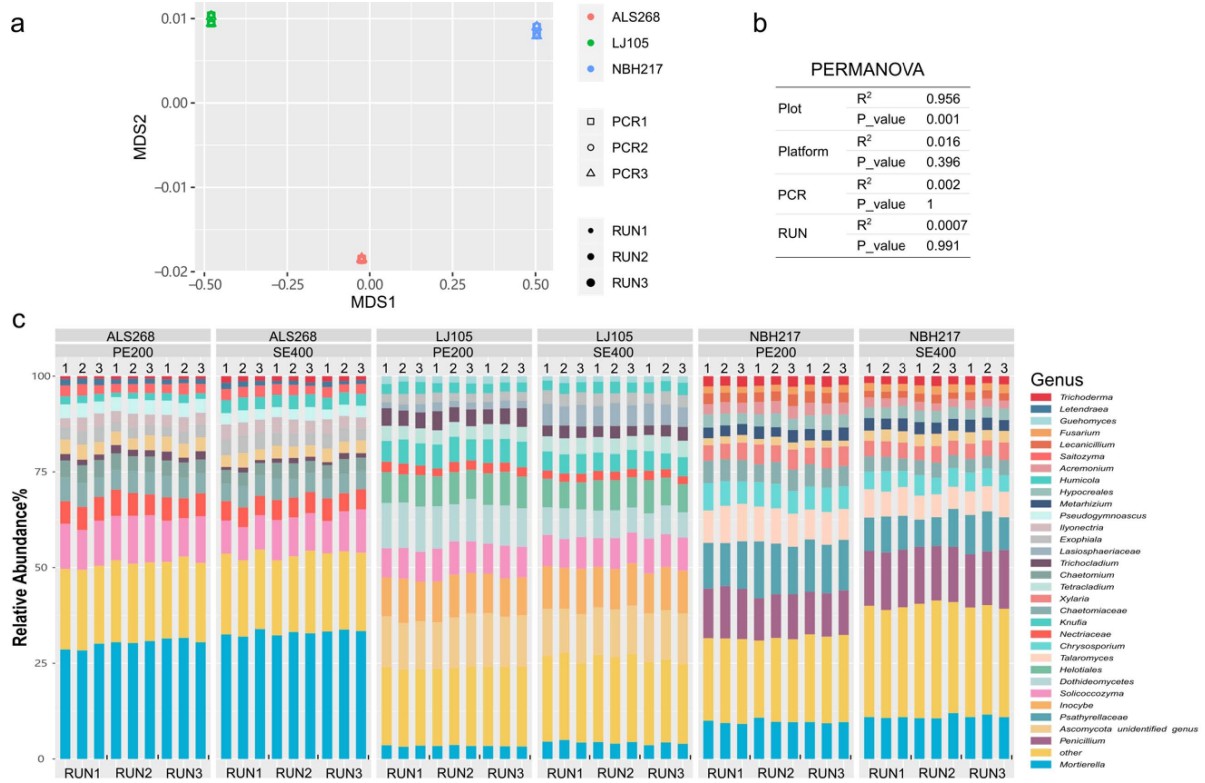

**Figure 5.** (a) Non-metric multidimensional scaling (NMDS) analysis and (b) Permutational multivariate analysis of variance (PERMANOVA) results showed no significant sequencing mode, run and PCR on the composition of fungal communities in three forest plots generated by DNBSEQ-G400 PE200 and SE400. (c) bar chart shows the percentage of major OTUs in three forest plots at genus level in DNBSEQ-G400 sequencer. ALS = AiLaoShan, LJ = LiJiang, NBH = NaBanHe.

machine. Consistent performance of DNBSEQ-G400 was observed. The average correlation coefficients of fungal composition from the three sequencing repeats were 0.96, 0.99 and 0.99 for ALS, LJ and NBH, respectively. Furthermore, the three replicates of all soil samples were significantly correlated (the average correlation coefficient was 0.98 for SE400_1 with SE400_2, 0.99 for SE400_1 with SE400_3, and 0.99 for SE400_2 with SE400_3; *P* < 0.001).

## Consistency of DNBSEQ-G400 evaluated by technical replicates

To evaluate the sequencing stability of the DNBSEQ-G400 platform, we performed NMDS ordination on the technical replicates of samples ALS268, LJ105 and NBH217, which were produced by three independent sequencing runs. Both PCR and run replicate analysis results were tightly clustered, and all technical replicates were separated according to their geological locations (Figure 5a). The observations were supported by higher $R^2$ value of samples organized by plot ($R^2$ = 0.956, *P* = 0.001) compared with samples organized by platform ($R^2$ = 0.016), PCR ($R^2$ = 0.002) and sequencing runs ($R^2$ = 0.0007) according to PERMANOVA (Figure 5b).

Technical replicates sequenced using two modes of DNBSEQ-G400 shared major fungal genera, but had varied relative abundance (Figure 5c). The differences in major fungal groups recovered by PE200 and SE400 were all less than 3%, except for *Dothideomycetes* in LJ, and *Psathyrellaceae* in NBH. Sequencing mode did not contribute significantly to the

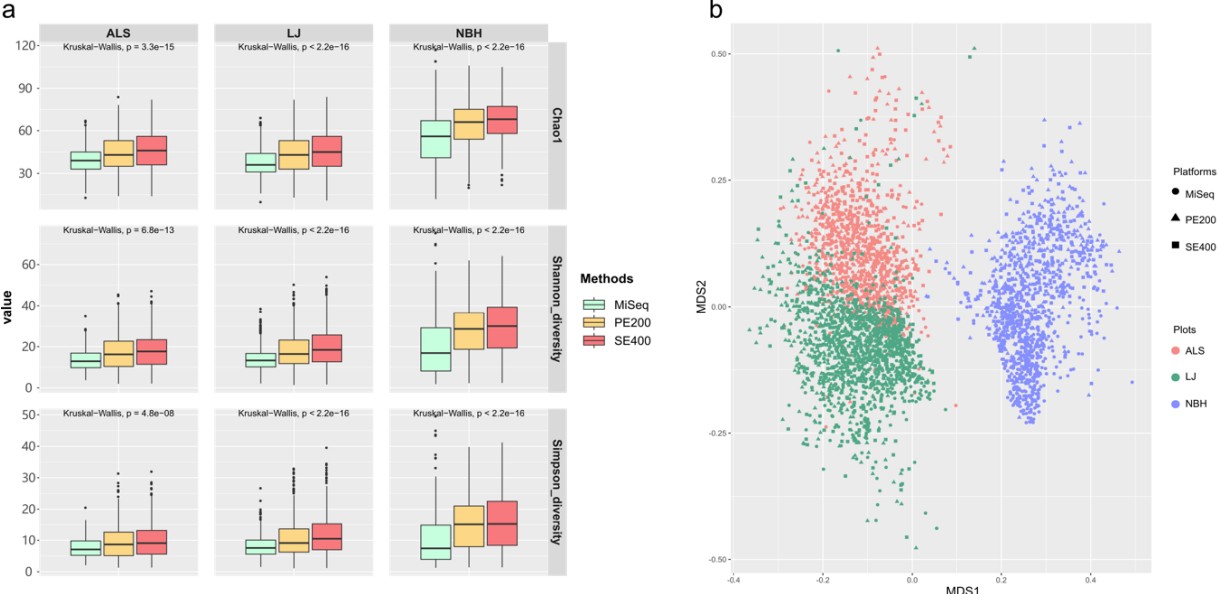

**Figure 6.** (a) Alpha-diversity was displayed by species richness (Chao1), Shannon and Simpson diversity index in all soil samples at different forest plots sequenced by MiSeq and DNBSEQ-G400. Data are visualized as box-and-whisker plots showing the median and the interquartile (midspread) range (boxes containing 50% of all values), the whiskers (representing the 25 and 75 percentiles) and the extreme data points. Numbers above boxes indicate significant differences (*p* value) by Kruskal-Wallis test. (b) Beta-diversity was displayed by NMDS of all soil species from 3 forest plots sequenced by MiSeq and DNBSEQ-G400.

observed β-diversity comparing to site (*P* = 0.77 for ALS, *P* = 0.28 for LJ, *P* = 0.56 for NBH by PERMANOVA) (Figure 5a, b).

## Taxonomic composition comparison between MiSeq and DNBSEQ-G400 with the same soil DNA samples

The ITS2 region of the same soil DNA samples were previously amplified with primer pairs 5.8SR_Nextera and ITS4_Nextera (dual-index [DI] approach) and were sequenced by Illumina MiSeq (unpublished). To evaluate the comparability and reuse availability of metabarcoding data from different sequencing platforms, we compared the two sequencing results in this study. Owing to a wider coverage of universal primers [42] in DNBSEQ, more species were recovered by DNBSEQ than MiSeq based on several alpha-diversity indexes (Figure 6a). Microbial profiles of samples from the same forest plots were clustered together by beta-diversity analysis (Figure 6b). A small but significant difference ($R^2$ = 0.0377) among sequencing platforms was observed by PERMANOVA (*P* = 0.001). Taxonomic characteriazation was affected by the selection of PCR amplification primers; this is consistent with previous reports [43, 44]. Since the amount of metabarcoding performed with DNBSEQ is likely to increase in the future, a comprehensive comparison of different sequencing platforms, such as the comparison described here, will provide useful information for design and standardization of experimental conditions.

## DISCUSSION

We comprehensively assessed the amplicon sequencing results of positive mock communities and a large number of soil samples using the DNBSEQ-G400 sequencing



platform. The DNBSEQ-G400 platform could restore the Zymo mock communities with negligible bias in our study (Figure 3a). High reproducibility of mock communities was also observed on MiSeq based on a previous study [45], showing an equivalent performance of the two sequencers. The substantial differences in quantification detected in an even mock community with seven fungi (Figure 3b) suggested sequencing bias against certain taxa. However, since this mock community sample was mixed based on genomic DNA content, rRNA copy number in genomic DNA might affect the detected taxonomic composition. In communities with different ratios of four fungi (Figure 3c), the abundance of *L. edodes* was overestimated in a combination of *L. edodes* and *S. cerevisiae.* The bias of relative proportion is best explained by the ITS1 length differences of the two species (*L. edodes* ITS1 is 243 bp, *S. cerevisiae* is 366 bp). Since DNBSEQ-G400 uses rolling-cycle amplification of sequencing signals, short fragments might be amplified more than long sequences in a given time, giving the impression of a higher abundance of *L. edodes.* This corresponds to much smaller biases in the abundance estimates of pairs of species with comparable ITS1 lengths: *F. velutipes* (247 bp) and *P. eryngii* (252 bp) relative to *L. edodes* (Figure 3c). This overestimation of taxa with short amplicons seems to be a general bias in metabarcoding; it was also observed as the main source of bias in observed community composition in other sequencers of Illumina and PacBio RSII and Sequel I [12].

Comprehensive comparisons of ITS2 amplicons from >1000 soil samples revealed highly comparable performance between the PE200 and SE400 modes of DNBSEQ-G400. Different fungal classes were confined to classes with very low abundance (Figure 4). A few profound differences between platforms at class-level abundances were detected; for example, the Archaeorhizomycetes class was highly detected by DNBSEQ PE200 and MiSeq in ALS and LJ, but not by SE400. Archaeorhizomycetes often have highly divergent sequences [42]; the missed detection in SE400 mode suggests the importance of 3′ sequences for assignment of certain taxa. Genera that were highly detected by SE400 modes but barely detected by PE200 were also poorly detected by PE200. Because we aligned R1 and R2 reads separately in paired sequencing modes, the read length for alignments is longest in SE400, and shorter in MiSeq, followed by PE200. Thus, differentially detected abundances in certain genera is probably because the read length affects the coverage of the recognition area during alignment. Taken together, these results suggest that care should be taken while jointly analyzing metabarcoding data from different sequencers or different sequencing modes.

A higher $R^2$ value was observed while evaluating run effects on technical replicates, suggesting that a large effect on taxonomic profile is caused by sequencing run (Figure 5b). As previous reports have showed Illumina sequencer run bias against certain fungal and archaeal taxa [17], technical controls must be included when executing large-scale sequencing efforts.

To date, comparable sequencing performance between DNBSEQ sequencers and Illumina sequencers has been reported in many studies, including palaeogenomic sequencing [21], metagenomics [24], and single-cell transcriptomics [25]. In contrast, significant differences in taxonomic composition based on short reads produced by MiSeq and DNBSEQ again indicated the potential problem of repeatability of metabarcoding caused by different primers and library preparation procedures. This has also been observed in some other studies [15, 18].

The rapid advances in sequencing technologies allows discovery of unknown microorganisms in natural complex samples, using techniques such as shotgun

metagenomics and long-read DNA sequencing methods [46–48]. However, it remains a difficult task for scientists to reflect the true taxonomic composition of a soil sample, owing to the largely undiscovered diversity; many undescribed species are genomically complex and have no reference sequences in the databases [49]. Compared with metagenomics, metabarcoding can provide sufficient taxonomic information in a more cost-effective manner, which makes it possible for researchers to expand their surveys of microbial communities to both large spatial and long temporal scales.

## REUSE POTENTIAL

Here, we evaluated the performance of the DNBSEQ-G400 platform for high-throughput metabarcoding in investigating the composition of soil microbial communities. The platform produced largely similar results as Illumina MiSeq (the currently dominant platform used for this purpose), but with a higher throughput and at potentially lower cost. The sequencing results of bacterial and fungal mock communities in this study could provide a reference for future evaluation stuides. Samples from three forest plots were from a long-term ongoing project. The fungal communities data can support further studies of the forest ecosystem in southwest China. The dataset in this study represents a substantial benchmark for opportunities and challenges presented by combining datasets produced by different sequencing platforms.

## DATA AVAILABILITY

Sequencing data for all samples have been deposited to the China National GeneBank DataBase (CNGBdb) [50], with the accession number CNP0000069, and the European Bioinformatics Institute (EBI) with the accession number PRJEB28018. All supporting data are deposited in the *GigaScience* GigaDB repository [27].

## DECLARATIONS
## LIST OF ABBREVIATIONS

ALS: AiLaoShan; cPAS: combinatorial Probe-Anchor Synthesis; DNB: DNA Nanoball; HTS: high-throughput sequencing; ITS, internal transcribed spacer; LJ: LiJiang; LSU, large subunit; NBH: NaBanHe; NMDS: Non-metric multidimensional scaling; OTU: operational taxonomic unit; PE200: paired-end 200bp; SE400: single-end 400bp.

## FUNDING

This research was supported by the National Natural Science Foundation of China (grant numbers 31570380, 31300358, 31100312), the West Light Foundation of the Chinese Academy of Sciences to Yue-Hua Hu, the CAS 135 program (grant number 2017XTBG-T01), the Natural Science Foundation of Yunnan Province (grant number 2015FB185), the Southeast Asia Biodiversity Research Institute, Chinese Academy of Sciences (grant number 2016CASSEABRIQG002), the National Key Basic Research Program of China (grant number 2014CB954100), and the Applied Fundamental Research Foundation of Yunnan Province (grant number 2014GA003).

## AUTHORS' CONTRIBUTIONS

X.S., L.X., Y.H. and Z.S. designed the study. Y.H. collected all soil samples from the three forests. X.S., H.M., H.Z., X.L. performed the amplification experiments. J.W., J.L., X.W. and

M.G. performed the DNBSEQ-G400 sequencing events. X.S., C.F., Y.J., analyzed the sequencing data. X.S., Y.H. and Z.S. wrote the manuscript.

## ACKNOWLEDGEMENTS

The authors are very grateful to colleagues at BGI-Shenzhen and China National Genebank (CNGB), Shenzhen for library constructions and discussions. We thank Professor Cene Gostinčar for helpful comments.

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
