## [Reviewer Report]

Comments on revised manuscriptI have now carefully read the revised version of this manuscript and I am happy with the changes that the authors implemented as a response to my comments and the comments of the other reviewer. The paper is now much more clear, especially in the methodological section and the limitations of the use of the novel sequencing platforms/formats is sufficiently discussed.

Minor comments that should be made in the present paper:

L58: change "bacteria" to "bacterial"
L65-66: the last part of this long sentence is difficult to comprehend and should be rephreased. I suggest to divide the long sentence into two
L68-69: change "produces" to "produced"
L84: delete "in"
L98: please explain the abbreviation "ONT", likely "Oxford Nanopore Technologies"
L162: the detail of the amplification methods should be expanded at least stating the primer pairs (names and sequences) used and targeted molecular markers; from the text it appears as if ITS2 was the marker selected, yet lines 361 and 366 discuss length differences in ITS1
L246: replace "common fungi several species" with "common fungal species"
L248-251: the misclassification of fungal taxa was not due to the bad performance of the sequencing platform, it was because of the low variability of the ITS2 marker. I suggest to change the text to state that genus level assignment was reached for these taxa since multiple species had the same ITS2 sequence
L264-265: the main reason is that the PCR bias (preferential PCR amplification of certain templates) skews the representation of taxa if the DNA is mixed prior to amplification
L331-346: this section is unclear; it should be specified which primers (primer names and sequences) with what barcodes were used for each conditions; if different primer pairs were used for different sequencing platforms, it is unclear what is the use of this comparison. This should be either clarified and explained all this section may be removed.
L381: delete "so"
L387-392: I suggest that this part is either removed or it is clearly described why the authors are sure that PCR replicates are not necessary (which is against all present recommendations). While the increasing fidelity of polymerases can be a fact, the main problems with parallel PCR is not errors (due to low fidelity) but random effects where primers align to templates with random frequencies. This statistical effect is impossible to handle by increasing polymerase fidelity while it is easily handled by PCR replication.
L424-426: This statement is rather obvious, I suggest to delete it.

---

## [Reviewer Report]

Reviewer name and names of any other individual's who aided in reviewer Inge SeimDo you understand and agree to our policy of having open and named reviews, and having your review included with the published papers. (If no, please inform the editor that you cannot review this manuscript.)YesIs the language of sufficient quality?NoPlease add additional comments on language quality to clarify if needed
The authors need to polish their English further. This is particularly obvious in the Abstract and is likely to result in an unwarranted lower readership of the work.Are all data available and do they match the descriptions in the paper? YesAdditional CommentsI want to commend the authors for sharing data and associated code.Are the data and metadata consistent with relevant minimum information or reporting standards? See GigaDB checklists for examples <a href="http://gigadb.org/site/guide" target="_blank">http://gigadb.org/site/guide</a>YesAdditional CommentsIs the data acquisition clear, complete and methodologically sound?YesAdditional CommentsIs there sufficient detail in the methods and data-processing steps to allow reproduction?YesAdditional CommentsIs there sufficient data validation and statistical analyses of data quality? Not my area of expertiseAdditional CommentsIs the validation suitable for this type of data?YesAdditional CommentsIs there sufficient information for others to reuse this dataset or integrate it with other data?YesAdditional CommentsAny Additional Overall Comments to the Author• R2 should be R^2 (that is, please superscript the '2').
• The sentence 'Further comparison between sequencing platforms would be useful for for exploration using as similar amplification conditions as possible. This data being provided as one such benchmark' at the end of Results is vague and needs to be rewritten.
• You need to more clearly state that you do not recommend to combine MGI and Illumina data sets for metabarcoding -- unlike e.g. BGISEQ-500 and Illumina RNA-seq/short-insert WGS data which can be readily combined.RecommendationMinor Revision

---

## [Reviewer Report]

Reviewer name and names of any other individual's who aided in reviewer Petr BaldrianDo you understand and agree to our policy of having open and named reviews, and having your review included with the published papers. (If no, please inform the editor that you cannot review this manuscript.)YesIs the language of sufficient quality?NoPlease add additional comments on language quality to clarify if needed
Are all data available and do they match the descriptions in the paper? NoAdditional CommentsI was not able to locate the items listed as references (26) and (27). Due to this, I was not able to fully evaluate the paper.Are the data and metadata consistent with relevant minimum information or reporting standards? See GigaDB checklists for examples <a href="http://gigadb.org/site/guide" target="_blank">http://gigadb.org/site/guide</a>NoAdditional CommentsI was not able to locate the data, see above.Is the data acquisition clear, complete and methodologically sound?NoAdditional CommentsMore details on sampling (mode of sampling, area sampled, depth sampled, sample size, sample handling) is missing. Information on number of repetitive extractions of DNA and the size of sample for extraction is missing. Protocols of amplification and barcoding are referenced as (27), but I was not able to locate this reference. These details have to be provided in the text for both types of sequencers.Is there sufficient detail in the methods and data-processing steps to allow reproduction?YesAdditional CommentsFor fungal ITS, the ITS region should be extracted before annotation.Is there sufficient data validation and statistical analyses of data quality? NoAdditional CommentsThe authors do not report how do they deal with sequences of fungi that produce amplicons longer than 350 bases that can not be pair-end joint in the 2x200 base runs. Even the MiSeq 2x250 runs miss some fungal taxa (though not very many) and here the situation is still worse. For the length distribution of fungal ITS, please consult the UNITE database.Is the validation suitable for this type of data?NoAdditional CommentsThere should be additional validations including the analysis of those OTUs that are abundant in one setup but missing in another one (if any).Is there sufficient information for others to reuse this dataset or integrate it with other data?NoAdditional CommentsThe metadata, supossedly in reference (26) are impossible to locate.Any Additional Overall Comments to the AuthorI believe that this is a very good attempt to test the novel platform with fungal metabarcoding. If all required information is provided, I believe that this can be both an interesting paper and a valuable dataset.RecommendationReject (Unsound or Unusuable)